# Knowledge, attitude and practices of community health workers on managing and preventing childhood malaria and diarrhea in Fako Division, South West Region, Cameroon; A mixed method study

**Ndum Naomi Chi**[1]*, **Raymond Babila Nyasa**[1,2], **Akoachere Jane-Francis**[1,3]

**1** Department of Microbiology and Parasitology, University of Buea, Buea, Cameroon, **2** Faculty of Science, Biotechnology Unit, University of Buea, Buea, South West Region, Cameroon, **3** Faculty of Science, Department of Microbiology and Parasitology, University of Buea, Buea, Cameroon

\* chinaomindum@gmail.com.

## Abstract

In developing countries, childhood malaria and diarrhea are among the 5 leading causes of death among children under five years; the use of community health workers (CHWs) to manage these diseases has shown some degree of success. A descriptive cross-sectional study was conducted to assess knowledge, attitude and practices (KAP) of CHWs on the management and prevention of childhood malaria and diarrhea in Fako Division, South-West Region, Cameroon. A pretested questionnaire was administered to eighty CHWs in Fako Division. Three focus group discussions (FGDs) were conducted with 29 CHWs. The Chi-Square and Spearman Correlation tests were used to investigate the association between socio-demographic characteristics with CHWs' KAP on childhood malaria and diarrhea management and prevention. A total of 52.5% of the participants had good knowledge, negative attitudes (65.0%), and carried out poor practices (60.0%) on the management and prevention of childhood malaria. Up to 8.75% CHWs did not know the first-line antimalarial drug used in Cameroon. More than half (57.5%) of participants had poor knowledge, 70.0% had a negative attitude and 82.25% carried out good practices on the management and prevention of childhood diarrhea. 47.6% of CHWs used a mixture of guava leaves and 'masepo' (*Ocimum gratissimum*) as treatment for childhood diarrhea. Level of education (p = 0.028) and Health District (p = 0.026) were significantly associated with practices on management and prevention of childhood diarrhea. CHWs had inadequate knowledge, poor attitude, and practices on childhood malaria management. Even though a majority of CHWs carried out good practices for diarrhea management, their knowledge and attitude were poor. Therefore, there is a need for training to improve CHWs' knowledge, attitude and practices on childhood malaria and diarrhea management.

**Data Availability Statement:** All relevant data are within the paper and its Supporting Information files.

**Funding:** The authors received no specific funding for this work.

**Competing interests:** The authors have declared that no competing interests exist.

**Abbreviations:** ACT, Artemisinin Combination Therapy; AIDS, Acquired Immunodeficiency Syndrome; CHW, Community Health Worker; FGD, Focus Group Discussions; HIV, Human Immunodeficiency Virus; KAP, Knowledge Attitude and Practice; LLIN, Long-lasting Insecticide-treated Net; NGO, Non-governmental Organization; ORS, Oral Rehydration Salt; RDT, Rapid Diagnostic Testing.

## Introduction

Malaria and childhood diarrhea are among the leading causes of morbidity and death worldwide. Young children and pregnant women are the most affected by malaria while children under the age of 5 years are the most affected by diarrhea [1,2]. In 2017, the global burden of malaria was estimated at 219 million and rose to 228 million in 2018 [3] with the WHO African region responsible for 93% of all malaria cases and 67% of global malaria deaths in children less than 5 years [4]. Based on country ranking, Cameroon is the world's 18th country with overwhelming <5 years mortality [5]. In 2018, Cameroon recorded more than 2 million cases of malaria and over 3000 deaths [6]. On the other hand, diarrhea is said to kill over 2,195 children each day, more than malaria, AIDS, and measles all combined [7]. In Cameroon, diarrhea is a major public health problem and 15% of deaths among children under 5 years of age are attributable to diarrheal diseases. The prevalence of Diarrhea in Tiko, a health district in the study area was 23.8% and children under 24 months were most affected [8].

The poorest and most disenfranchised populations in sub-Saharan Africa have limited access to quality healthcare facilities and services [9]. The healthcare worker-to-population ratio in Cameroon is estimated to be between 0.5 and 1 per 1000 [10]. The ratio of CHW to population in Fako Division (South-West Region) is 1:3740 and the health facility to population ratio is 1: 11,627. These are values observed during the fieldwork of this study since there is no available information on published data [10]. As a result, most countries in this region rely on CHWs to bring healthcare services and interventions to hard-to-reach and underserved populations to manage these maladies [11]. In Cameroon like in many endemic countries, the public health system uses CHWs to respond to the needs of these underserved populations [12]. CHWs have the potential to reduce mortality by improving access to care as they often receive training on health interventions to carry out defined functions related to healthcare delivery. Thus, they are used to improve community health initiatives, manage the risk of infectious diseases (e.g. diarrhea, malaria, and pneumonia), and fill gaps in healthcare systems [12–14]. The criteria for selection of CHWs is based on the country's context, the population's health needs and priorities, and the selection is done by local community members [12]. CHWs carry out activities such as home visits, provision of first aid and treatment of simple and common ailments, provision of health education on maternal and child health, family planning, TB and HIV/AIDS care, malaria control, communicable disease control, and other activities such as collection of data on vital events [13,15].

Recent studies have shown that community and facility healthcare workers have inadequate knowledge and do not adhere to diarrhea and malaria case management guidelines, which could be contributing to increased morbidity and mortality [16,17]. Likewise, other studies have reported misconceptions about malaria transmission among primary caregivers indicating remaining knowledge gaps in educational campaigns [18,19]. Recent studies have reported associations between socio-demographic factors (such as being single) and CHWs' knowledge on malaria management and control [19]. Although much research has been done on childhood diarrhea and malaria in Kenya, Zambia, Rwanda, Uganda, Somaliland, Cameroon, Nigeria, and Senegal [16–24], few studies have evaluated CHWs' KAP on the management of malaria and diarrhea in Cameroon [19], especially as many cases still report in hospitals in the study area [25,26]. It is probable that CHWs have low knowledge, negative attitudes, and carry out poor practices on the management of these diseases. This study aimed at evaluating CHWs KAP on childhood diarrhea and malaria management in Fako Division (Limbe, Tiko, Buea).

## Materials and methods

### Study setting

The study was carried out in Fako, one of the six divisions in the South-West Region of Cameroon. Fako Division is divided into 5 Sub-Divisions Buea, Muyuka, Tiko, Limbe, and Idenau. With regards to its health structure, Fako Division consists of four health districts; Tiko, Buea, Limbe, and Muyuka. Each health district is subdivided into health areas. Buea health district has 7 health areas; Buea Town, Bova, Muea, Molyko, Bokwango, Tole, and Soppo. There are 8 health areas in the Limbe health district; Bota, Bojongo Bonadikombo, Limbe Zone II, Idenau Moliwe, Limbe Town, and Limbe Sea Port; 8 health areas in the Muyuka health district: Owe, Ekata Bafia, Muyenge, Yoke, Malende, Meanja and Mpundo, while Likomba, Mondoni, Missellele, Mutengene, Molewe, and Tiko Town are the 6 health areas in Tiko health district. Fako Division has an estimated population of 534,854 inhabitants [27]. The climate of Fako Division is hot and dry. There are two seasons; the dry season and a rainy season with an annual rainfall of about 1500 mm/year inland and 10,000 mm/year on the seacoast [28]. Fako Division is surrounded by the humid savannah and deep equatorial evergreen rainforest and vegetation. Agriculture and fishing are the major economic activities of the people. A large proportion of the population works at the Cameroon Development Cooperation, while part of the population is also made up of civil servants, business people and students.

The presence of CHWs in the healthcare system has led to increased access to and utilization of services by populations in hard-to-reach areas. From 2014 to 2016 during periods of malaria sweeps in Senegal, the involvement of CHWs in passive case detection in villages led to a 104% increase in rapid diagnostic test (RDT) testing, 77% of positive malaria cases detected and treated using artemisinin-based combination therapy (ACT) [24]. In Cameroon, CHWs are trained to contribute to the delivery of both primary and preventive care elements of primary health care services in the communities they serve. They develop trusting, one-on-one relationships with community members and have the potential to facilitate improvements in health status and quality of life in rural communities since they are familiar with the beliefs, culture, and language of the local community and are always available at the community level to offer timely healthcare services. The CHWs who participated in this study worked with both public and private health facilities within the district. Fifteen health facilities from Limbe, 14 from Tiko and 17 from Buea.

### Study design

This was a community-based cross-sectional mixed method study conducted among CHWs in Fako Division, South-West Region, Cameroon. This study was limited to assessing CHWs' KAP on the management of malaria and childhood diarrhea. A list of CHWs in each health area and chiefs of center (health center managers) were obtained from the various health districts. With the assistance of the various chiefs of center, all CHWs in the health areas were contacted through phone calls to participate in the study. CHWs recruited in this study were exhaustively selected from 3 of the four health districts: Buea (Buea Town, Bova, Muea, Molyko, Bokwango and Soppo), Limbe (Bota, Bonadikombo, Limbe Zone II, Moliwe, Limbe Town and Limbe Sea Port), and Tiko (Likomba, Missellele, Mutengene, and Tiko Town) from January to September 2020. Muyuka health district was left out due to insecurity resulting from the present sociopolitical crisis in the country, which is aggravated in the region as an armed conflict.

Out of the 143 CHWs in Fako Division, 80 took part in this study. A list of all recruited CHWs was obtained at the district level with their contacts. The CHWs were invited to a health

facility in the district, where the researchers administered the questionnaire. For the FGDs, 29 CHWs were conveniently selected to participate and they included; 9 CHWs from Tiko, 11 from Buea, and 9 from Limbe health districts. The FGDs were conducted in English language facilitated by the researcher. During the FGDs, the researcher trained the CHWs on their roles in the management of these diseases.

### Ethical considerations

Ethical approval for this study was obtained from the Faculty of Health Sciences Institutional Review Board of the University of Buea (reference number 2020/1202-04/UB/SG/IRB/FHS). Administrative authorizations were obtained from the Regional Delegation of Public Health for the South-West Region (reference number R11/MINSANTE/SWR/RDPH/PS/968/914) and from the Buea, Limbe, and Tiko health district services. Participation in the study was voluntary and written informed consent was obtained from all participants before their recruitment into the study. Questionnaires were coded to ensure confidentiality and participants were educated on childhood malaria and diarrhea management at the end of the study.

### Questionnaire administration

The questionnaire multiple-choice questions (S1 Questionnaire) was pre-tested on 15 CHWs in Kumba sub-division to ascertain validity and clarity. The questionnaire was adapted from previous studies and composed of four sections [18–20].

Section 1 contained questions on participants' socio-demographic characteristics; age, sex, marital status, the highest level of education, religion, longevity in service as a CHW, occupation and the health district. Questions on the presence and number of children below 5 years old were used to ensure their presence in the community hence their availability to care by CHW. Section 2 captured information on CHW's knowledge on the management of children with malaria and diarrhea (12 and 10 questions respectively). This included the definition of malaria, signs and symptoms of malaria, treatment and prevention of malaria, the definition of diarrhea, recognition of signs and symptoms of diarrhea and treatment options. Section 3 was on CHW's attitudes (beliefs) towards the management of childhood malaria and diarrhea (15 and 9 questions respectively). This included closed-ended questions on the cause of diarrhea, the use of leaves for malaria treatment, malaria prevention, malaria as a life-threatening disease, malaria as a disease of the poor, diarrhea infection can lead to death. Section 4 captured information on CHWs practices in managing children with malaria and diarrhea (12 and 17 questions respectively). The questions assessed their use of oral rehydration salts (ORS) solution, malaria treatment administered, and their participation in the distribution of long-lasting insecticide-treated nets (LLINs)

### Criteria for inclusion/exclusion

The study included all CHWs aged 21 years and above who had served the community for at least 1 month and were residents in Fako Division. CHWs who refused to give their consent were not allowed to participate in the study. The FGDs were composed of 29 conveniently selected CHWs who already completed the questionnaire.

### Focus group discussion

FGDs were conducted on topics related to childhood malaria and diarrhea management and prevention; the definition of childhood malaria and diarrhea, causes, signs and symptoms, prevention and treatment options (S1 Text). The discussions were recorded using a tape recorder.

### Data analysis

The KAP survey data survey (S1 Data) was entered into Epi Info database and analyzed using Statistical Package for the Social Sciences (SPSS) version 25 [29]. Continuous variables (age, number of children less than 5 years, years of experience) were described using frequency, means and standard deviations. Frequencies and percentages were used to describe categorical variables (knowledge, attitude and practices). Each question was given a score of zero or one for the wrong or right answer respectively. Knowledge, attitude and practices scores were each calculated by summing scores for all the questions under each section. CHWs' level of KAP were determined by considering any value below the mean as poor or negative, while any value above the mean was considered good or positive [19]. Knowledge of malaria and diarrhea were scored on 16 and 30 respectively, attitude on 18 and 11 while practices were on 14 and 38. The Chi-square test was used to investigate the association of socio-demographic factors with CHWs' KAP on the management and prevention of childhood malaria and diarrhea. The analysis considered CHWs' KAP as binary outcome variables and socio-demographic factors as predictors. Spearman rank correlation was used to investigate the association of CHWs' knowledge score with attitudes and practices on the management of childhood malaria and diarrhea. This research study considered 95% confidence intervals and a P-value <0.05 for significance.

Data from the FGDs were analyzed by two of the authors who used reflective thematic analysis (Table A in S1 Table). Data from the recorded FGDs were transcribed into Microsoft Word 2010. The comments for all the FGDs were grouped to capture the main idea of participants' responses to each question. Coding was done by considering responses with common main ideas. Using critical thinking, recurrent codes were identified and grouped under common themes. Further critical thinking was done on the common themes to generate the main themes which were reported as the research findings [30].

## Results

The general characteristics of the participants are shown on Table 1. The mean age (SD) of the 80 CHWs was 47.7 years (12.164).

The FGDs in Buea, Tiko and Limbe were composed of 9, 9, and 11 participants respectively (Table 2). A greater proportion of these participants had served as CHWs for more than 5 years (58.6%) and the mean age for study participants was 47.7 years with a standard deviation of 12.164.

### Knowledge on management and prevention of childhood malaria

Fifty (62.5%) CHWs reported malaria is caused by the female *Anopheles* mosquito while 10 (12.5%) attributed it to poor hygiene and sanitation (Table 3). The majority (76, 95.0%) reported malaria is transmitted through a mosquito bite and that this mosquito bites mostly at night 45(56.25%). Fifty-two (65.0%) participants correctly identified the signs and symptoms of simple malaria to be "Fever, headache, loss of appetite and body pains", only eight (10.0%) reported loss of consciousness with temperature >37˚C as signs and symptoms of severe malaria.

Concerning the management of a suspected case of simple malaria, only 24 (30.0%) of CHWs strongly agreed that a RDT and treatment with anti-malarial medication should be done for every suspected case of malaria slightly while above half (52.5%) recommended referral to the hospital for care. Only thirty-four respondents (42.5%) knew correctly that ACT is the first-line treatment for malaria in Cameroon. Other participants indicated quinine sulphate (30.0%), amodiaquine (13.75%) and chloroquine (5.0%). Regarding the duration of

**Table 1. Socio-demographic characteristics of participants (n = 80).**

| Variable | Frequency (n) | Percentage (%) |
|---|---|---|
| **Gender** | | |
| Male | 32 | 40.0 |
| Female | 48 | 60.0 |
| **Age group(years)** | | |
| 22–30 | 14 | 17.5 |
| 31–40 | 26 | 32.5 |
| 41–50 | 15 | 18.8 |
| 51–60 | 19 | 23.8 |
| >60 | 6 | 7.5 |
| **Have a child of <5 years** | | |
| No | 39 | 48.8 |
| Yes | 41 | 51.3 |
| **Number of children <5 years** | | |
| 0 | 38 | 47.5 |
| 1 | 22 | 27.5 |
| 2 | 13 | 16.3 |
| >2 | 7 | 8.7 |
| **Marital status** | | |
| Single | 32 | 40.0 |
| Married | 33 | 41.3 |
| **Highest level of education** | | |
| No education | 1 | 1.25 |
| Primary | 22 | 27.5 |
| Secondary | 48 | 60.0 |
| University | 9 | 11.25 |
| **Other Occupation** | | |
| Unemployed | 54 | 67.5 |
| Employed (having another job aside from being a CHW) | 26 | 325 |
| **Health District** | | |
| Buea | 18 | 22.5 |
| Tiko | 20 | 27.5 |
| Limbe | 40 | 50.0 |

treatment, the majority of participants (71.25%) reported that treatment for malaria should be administered for 3 days and 10 (12.5%) reported five days. The majority of CHWs (97.5%) advised community members to wash, air-dry and reuse mosquito nets as preventive measures for malaria.

Participants of the FGD knew that malaria is caused by an infected female *Anopheles* mosquito, while others however reported the mosquito as seen below:

> *"Infected female Anopheles mosquito."* [Female, LIMBE]

> *"Malaria is caused by mosquitoes."* [Male, TIKO]

> *"Female Anopheles mosquito."* [Male, BUEA]

Some of the FGD participants reported having been trained on the management of simple malaria.

**Table 2. Socio-demographic characteristics of participants of FGDs (n = 29).**

| Variable | Frequency (n) | Percentage (%) |
|---|---|---|
| **Gender** | | |
| Male | 13 | 44.8 |
| Female | 16 | 55.2 |
| **Health District** | | |
| Buea | 9 | 31.03 |
| Limbe | 11 | 37.9 |
| Tiko | 9 | 31.03 |
| **Age Range** | | |
| Minimum | 26 | |
| Maximum | 67 | |
| **Years of experience (years)** | | |
| 0–5 | 12 | 41.4 |
| 5.1–10 | 11 | 37.9 |
| 10.1–15 | 3 | 10.3 |
| >15 | 3 | 10.3 |

*"Some (a few) of us were selected and given training on how to manage malaria. We were taught how to do RDT and were told that we will be given test strips, drugs and other items which we could use in the community to manage malaria. Until today, nothing has been given to us. It's been more than five years now; we are still waiting. Yes, we have attended seminars about malaria."* [Female, TIKO]

*"Management of malaria and childhood diarrhea is not in our package of activities as CHWs."* [Male, TIKO]

The FGD participants reported signs and symptoms of malaria including headache, joint paints, fever, and loss of appetite. For the first-line treatment for malaria in Cameroon, most of them correctly stated artemether-lumefantrine and artesunate-amodiaquine while others reported the use of Quinine Sulphate, as common anti-malaria drugs used in their communities.

*The signs and symptoms of malaria are fever, body weakness, and broken joint pains. [Female, TIKO]*

*"Artemether is the first line treatment for malaria."* [Male, BUEA]

*"Quinine Sulphate is used in treating malaria."* [Male, LIMBE]

*"Bimalaril is one of the antimalarial drugs we use."* [Female, LIMBE]

Concerning methods of malaria prevention, FGD participants knew that malaria can be prevented by sleeping under LLINs, clearing bushes around the house, emptying potholes, wearing long sleeve clothes and by the use of insecticides.

*"Sleeping under insecticide-treated mosquito nets."* [Female, LIMBE]

*"Wear long sleeve clothes, and clear bushes around the house."* [Female, BUEA]

*"Emptying potholes around the house."* [Male, LIMBE]

**Table 3. CHWs' Knowledge of management and prevention of childhood malaria.**

| Variable | Frequency (n) | Percentage (%) |
|---|---|---|
| **Cause of malaria** | | |
| Not using a mosquito net | 15 | 18.8 |
| Housefly or mosquito | 1 | 1.3 |
| Poor hygiene and sanitation | 10 | 12.5 |
| Female *Anopheles* mosquito | 50 | 62.5 |
| Mosquito or a tsetse fly | 4 | 5.0 |
| Blood transfusion | 0 | 0 |
| **Malaria transmission** | | |
| Insect bite | 2 | 2.5 |
| Wearing dirty clothes | 0 | 0 |
| Mosquito bite | 76 | 95.0 |
| Drinking dirty water | 2 | 2.5 |
| **The insect that transmits malaria bite most during** | | |
| Day time | 1 | 1.3 |
| Night time | 45 | 56.3 |
| Both during the day and at night | 34 | 42.5 |
| I do not know | 0 | 0 |
| **Signs and Symptoms of simple malaria** | | |
| Loss of consciousness, headache, loss of appetite, joint pains | 15 | 18.8 |
| Fever, loss of consciousness, headache and body pains | 12 | 15.0 |
| Fever, headache, loss of appetite and body pains | 52 | 65.0 |
| Vomiting, fever and watery stool | 1 | 1.3 |
| **Signs and symptoms for severe malaria** | | |
| Vomiting within 24 hours | 9 | 11.3 |
| Loss of consciousness with temperature >37˚C | 8 | 10.0 |
| Difficulties in breathing with temperature>37˚C | 55 | 68.8 |
| Lack of appetite | 8 | 10.0 |
| **Every suspected malaria case should do an RDT** | | |
| I agree | 55 | 68.8 |
| I do not agree | 9 | 11.3 |
| I do not know | 16 | 20.0 |
| **How to manage a patient with signs and symptoms of simple malaria** | | |
| Refer to the hospital or health center | 42 | 52.5 |
| Give paracetamol to calm the fever | 14 | 17.5 |
| Allow the patient to rest | 0 | 0 |
| Do RDT and treat with anti-malaria drugs | 24 | 30.0 |
| **First-line antimalaria drugs in Cameroon** | | |
| Chloroquine | 4 | 5.0 |
| Artemisinin combination therapy (ACT) | 34 | 42.5 |
| Amodiaquine | 11 | 13.8 |
| Quinine Sulphate | 24 | 30.0 |
| I do not know | 7 | 8.8 |
| **Duration of treatment for simple malaria** | | |
| 1 day | 6 | 7.5 |
| 3 days | 57 | 71.3 |
| 5 days | 10 | 12.5 |
| I week | 7 | 8.8 |

(*Continued*)

**Table 3.** (Continued)

| Variable | Frequency (n) | Percentage (%) |
|---|---|---|
| I do not know | 0 | 0 |
| **Advice about mosquito net** | | |
| Throw it away after 6 months or when it gets dirty | 1 | 1.3 |
| Mosquito nets are reusable; wash, air-dry then reuse | 78 | 97.5 |
| Unmount and never reuse mosquito nets | 1 | 1.3 |
| No advice | 0 | 0 |
| I do not know | 0 | 0 |

## Attitude towards childhood malaria management and prevention

Only 36 (45.0%) of CHWs believed that mosquitos that transmit malaria get the infection by biting an infected individual, and all of them (100.0%) agreed that sleeping under a mosquito net prevents malaria (Table 4). However, 7.5% agreed that severe malaria can be caused by witchcraft. Forty-three (53.75%) agreed that every child <5 years old should routinely be screened for malaria. Sixty (75.0%) CHWs agreed that a mixture of boiled paw-paw, mango leaves and fever grass (lemon grass) is a good home treatment for malaria. Thirty-nine (48.75%) of the respondents reported: "children <5 years and pregnant women" as those most susceptible to malaria. In addition, 97.5% regarded malaria as a life-threatening illness and 54 (67.5%) were in disagreement that, the poor are more likely to be infected with malaria. Community members obtained malaria treatment from multiple sources; traditional healers (66, 82.5%), CHW (40, 50.0%), street vendors (15, 18.75%), hospitals (46, 57.5%) and none from the pharmacy, while 8 (10.0%) did not know. Half of the participants (40; 50%) reported that the use of LLINs produce heat. Sixty-four (80.0%) participants recommended RDT and treatment of every suspected malaria case. However, the majority of participants (76, 95.0%) viewed malaria as a serious health problem in their community.

Some participants of the FGDs used a concoction of boiled leaves as treatment for childhood malaria:

> "In my case, I do not refer the patient immediately. Usually, I encourage the use of traditional herbs. A mixture of paw-paw leaves, guava leaves mango leaves, 'masepo', fever grass and others which I boil very well and then, drink from a teacup, I also cover myself with it every morning and evening. I do this for three days. On the third day of the treatment, I heat the remaining treatment and then use it to insert it into the child's anus, to wash the stomach. I do this for the kids and also for myself when I have malaria. If malaria persists, I then refer the child to the hospital." [Female, LIMBE].

## Practices regarding management and prevention of childhood malaria

Sixty-seven (83.75%) CHW had managed at least one malaria case. Amongst them 75.0% administered treatment and only 42 (57.5%) did RDT (Table 5). Fifty-one (63.75%) CHWs always participated in the distribution of mosquito nets and 8 (10%) had never taken part in this activity. Above half of the respondents (51.25%) had never given health education on hygiene and sanitation to community members. Sixty-eight (85.0%) always prescribed anti-malaria drugs. Most respondents (82.5%) did not have mosquito nets for distribution at the time of the study and 43 (53.8%) preferred the use of both LLINs and insecticide spray as malaria preventive measures. For treatment with anti-malarial drugs, more than half of the

**Table 4. CHWs' attitude on childhood malaria management and prevention.**

| Variable | Frequency (n) | Percentage (%) |
|---|---|---|
| **Insects that cause malaria get the infection from** | | |
| Biting individuals infected with malaria | 36 | 45.0 |
| Contaminated water | 21 | 26.3 |
| From the atmosphere | 3 | 3.8 |
| All mosquitoes are born already infected | 18 | 22.5 |
| I don't know | 2 | 2.5 |
| **Sleeping under mosquito nets can prevent malaria** | | |
| I agree | 80 | 100.0 |
| I do not agree | 0 | 0 |
| I do not know | 0 | 0 |
| **Severe malaria can be caused by witchcraft** | | |
| I agree | 6 | 7.5 |
| I do not agree | 70 | 87.5 |
| I do not know | 4 | 5.0 |
| **A mixture of boiled paw-paw, mango leaves, fever grass is a good home treatment for malaria** | | |
| I agree | 60 | 75.0 |
| I do not agree | 16 | 20.0 |
| I do not know | 4 | 5.0 |
| **Every child < 5 years should routinely be screened for malaria** | | |
| I agree | 43 | 53.8 |
| I do not agree | 28 | 35.0 |
| I do not know | 9 | 11.3 |
| **Which group is most susceptible to malaria** | | |
| Adults | 2 | 2.5 |
| Children < 5 years | 5 | 6.3 |
| Children <5 years and pregnant women | 39 | 48.8 |
| Pregnant women | 0 | 0 |
| Everyone | 34 | 42.5 |
| I do not know | 0 | 0 |
| **Malaria is a life-threatening illness** | | |
| Yes | 78 | 97.5 |
| No | 2 | 2.5 |
| I do not know | 0 | 0 |
| **The poor are more likely to be infected with malaria** | | |
| I strongly agree | 7 | 8.8 |
| I agree | 14 | 17.5 |
| I do not agree | 54 | 67.5 |
| I do not know | 5 | 6.3 |
| **Where do community members get treatment for malaria** | | |
| Traditional healers | 66 | 82.5 |
| CHWs | 40 | 50.0 |
| Street drug vendors | 15 | 18.8 |
| Hospital | 46 | 57.5 |
| Pharmacy | 0 | 0 |
| I do not know | 8 | 10.0 |
| **Mosquito net makes places to be hot** | | |

*(Continued)*

**Table 4.** (Continued)

| Variable | Frequency (n) | Percentage (%) |
|---|---|---|
| I agree | 40 | 50.0 |
| I do not agree | 38 | 47.5 |
| I do not know | 2 | 2.5 |
| **RDT and treatment are recommended for every malaria case** | | |
| I agree | 64 | 80.0 |
| I do not agree | 10 | 12.5 |
| I do not know | 6 | 7.5 |
| **Malaria is a serious health problem in my community** | | |
| I agree | 76 | 95.0 |
| I do not agree | 3 | 3.3 |
| I do not know | 1 | 1.3 |
| **Side pain is a cause of malaria in children** | | |
| I agree | 37 | 46.3 |
| I do not agree | 30 | 37.5 |
| I do not know | 13 | 16.3 |

CHWs (53.75%) administered 2 doses/day, while 13 (16.25%) did not know the number of treatment doses to administer in a day.

Regarding what is done when a case of malaria is identified, FGD participants reported that they immediately refer the patient to the nearest hospital or health center, a few prescribed antimalarial medications while very few said they do an RDT. However, some of the participants reported that they use concoctions made of boiled leaves and tree backs and administer a full glass to malaria patients. After which, they cover the patient with a thick blanket to inhale the vapor.

*"I do not refer the patient immediately. Usually, I encourage the use of traditional herbs; a mixture of paw-paw leaves, guava leaves mango leaves, 'masepo' (Ocimum gratissimum), fever grass (Cymbopogon citratus) and others which I boil very well then, give the patient to drink from a teacup. The patient also does steam inhalation of the concoction every morning and evening for three days. On the third day, I heat the remaining treatment and then use it to "pump" the patient. If malaria persists, I then refer the patient to the hospital."* [Male, LIMBE]

*"With severe malaria, I use sand leaves, paw-paw leaves, fever grass with lemon and blackjack (Bidens Pilosa) which I boil, sieve and put in 5liter containers. I give the patient one glass of the mixture three times a day for seven days. Usually, I encourage the parents to use a blanket and cover the child above the heat coming from the pot of the boiled mixture."* [Female, BUEA]

*"I mix paw-paw leaves, lemon, blackjack, back of mango tree, back of pear tree and boil. After it cools, I sieve and put in 1.5 liters containers, then I give one glass of the mixture to the patient in the morning and another glass in the evening for five days. Sometimes I encourage the parent to cover the child with a thick cloth/blanket over the pot of boiled mixture to hasten the healing process."* [Male, BUEA]

*"I do the RDT on a suspected malaria patient and if he or she tests positive I then give an antimalarial medication."* [Female, TIKO]

**Table 5. CHWs' practices on management and prevention of childhood malaria.**

| Variable | Frequency (n) | Percentage (%) |
|---|---|---|
| **Ever managed a case of malaria** | | |
| Yes | 67 | 83.8 |
| No | 13 | 16.3 |
| **Have you ever performed a rapid diagnostic test** | | |
| Yes | 42 | 57.5 |
| No | 38 | 47.5 |
| **Have you ever administered treatment for simple malaria?** | 60 | 75.0 |
| Yes | 20 | 25.0 |
| No | | |
| **Participation in the distribution of mosquito nets** | | |
| Always | 51 | 63.8 |
| Sometimes | 20 | 25.0 |
| Never | 8 | 10.0 |
| When I have the time | 0 | 0 |
| I do not know | 1 | 1.3 |
| **Give health education on hygiene and sanitation** | | |
| Always | 12 | 15.0 |
| Sometimes | 27 | 33.8 |
| Never | 41 | 51.3 |
| I do not know | 0 | 0 |
| **Prescribe malaria drugs** | | |
| Always | 68 | 85.0 |
| Sometimes | 12 | 15.0 |
| Never | 0 | 0 |
| I do not know | 0 | 0 |
| **Have mosquito nets for distribution** | | |
| Yes | 14 | 17.5 |
| No | 66 | 82.5 |
| **Which malaria preventive measure do you prefer** | | |
| Clearing bushes around the farm | 6 | 7.5 |
| Use insecticide sprays | 1 | 1.3 |
| Use mosquito nets | 30 | 37.5 |
| Use insecticide spray and mosquito nets | 43 | 53.8 |
| **Doses of antimalarial drugs administered in a day to a child** | | |
| Single | 19 | 23.8 |
| Two | 43 | 53.8 |
| Three | 5 | 6.3 |
| Five | 0 | 0 |
| I do not know | 13 | 16.3 |

*"I just refer the patient to the hospital."* [Male, LIMBE]

*"After our training, we used to be given Artesunate amodiaquine (AZAC) to manage malaria in the community, but for about one year now we have not received these drugs from the Drug Fund."* [Female, BUEA]

*"I work with Relief International where they give me drugs and all materials to manage malaria in the community."* [Male, BUEA]

*"I prescribe anti-malaria drugs to some and refer cases which I consider serious to the hospital."* [Female, LIMBE]

## Correlation of CHWs scores on KAP on childhood malaria management and prevention

KAP scores on childhood malaria management and prevention from the questionnaire were calculated by summing the total score for all the questions under each section, with 16 as the total score for knowledge, 18 for attitude and 14 for practices (Table 6). The mean score (± SD) for knowledge of childhood malaria management and prevention was 11.48 (±3.49), attitude 7.98 (±2.63) and practices 7.01 (±2.18). Bivariate correlation showed that knowledge had a significant positive correlation with attitude (r = 0.383, P-value = 0.0001) and attitude had a significant positive correlation with practices (r = 0.257, P-value = 0.010).

The Chi-square test indicated no significant association between socio-demographic factors and participants' KAP on childhood malaria management and prevention. More than half of the CHWs had good knowledge (52.5%), negative attitude (65.0%), and carried out poor practices (60.0%) on the management and prevention of childhood malaria (Fig 1).

## Knowledge on childhood diarrhea management and prevention

The majority of participants 69(86.25%) correctly defined diarrhea as "Passage of loose or watery stool more than 3 times a day" (Table 7). A total of 72 (90.0%) of them identified "Unsafe drinking water", "teething" (15%), and "intravenous use of drugs" (2.5%) as the cause of childhood diarrhea. Signs and symptoms of diarrhea identified were "weakness" (46.38%), "watery stool greater than four days" (31.35%), "weight loss" (25%), "blood in watery stool" (21.3%), "sunken eyes" (20%). Eighteen (22.5%) were in agreement that abstaining from breast milk prevents childhood diarrhea. Majority of the participants (93.75%) knew how to prepare ORS, and most (60%) always use ORS as a treatment for childhood diarrhea. However, 28.8% did not know that zinc supplements are used in treating childhood diarrhea. Fifty-five (68.8%) participants identified "weight loss" as one of the consequences of diarrhea while only 13 (16.3%) were aware that diarrhea in children could result in death. The majority of participants advised community members to seek help for the management of diarrhea in children from health centers 66(82.5%) and hospitals 62 (77.5%).

Participants of FGDs reported signs and symptoms of childhood diarrhea to include; body weakness, frequent stools >3 times a day, loss of appetite, fever, dehydration, weight loss and sunken eyes. They gave a wide variety of definitions for diarrhea. Some defined diarrhea

**Table 6. Descriptive statistics for KAPs of CHWs regarding the management of childhood malaria.**

| Variable | Total | Minimum | Maximum | Mean | Standard deviation | Spearman Correlation and $X^2$ Value | | |
|---|---|---|---|---|---|---|---|---|
| | | | | | | Knowledge | Attitude | Practices |
| **Knowledge Score for malaria management** | 16 | 4 | 16 | 11.48 | 3.49 | r = 1.00 | r = 0.383** | r = 0.010 |
| **Attitude Score for malaria management** | 18 | 2 | 13 | 7.98 | 2.629 | r = 0.383** p-value = 0.0001 | r = 1.000 | r = 0.257* |
| **Practices Score for malaria management** | 14 | 1 | 12 | 7.01 | 2.179 | r = 0.010 | r = 0.257* P-value = 0.010 | r = 1.000 |

** Correlation is significant at the 0, 01 level (two-tailed).

*Correlation is significant at the 0.05 level (two-tailed).

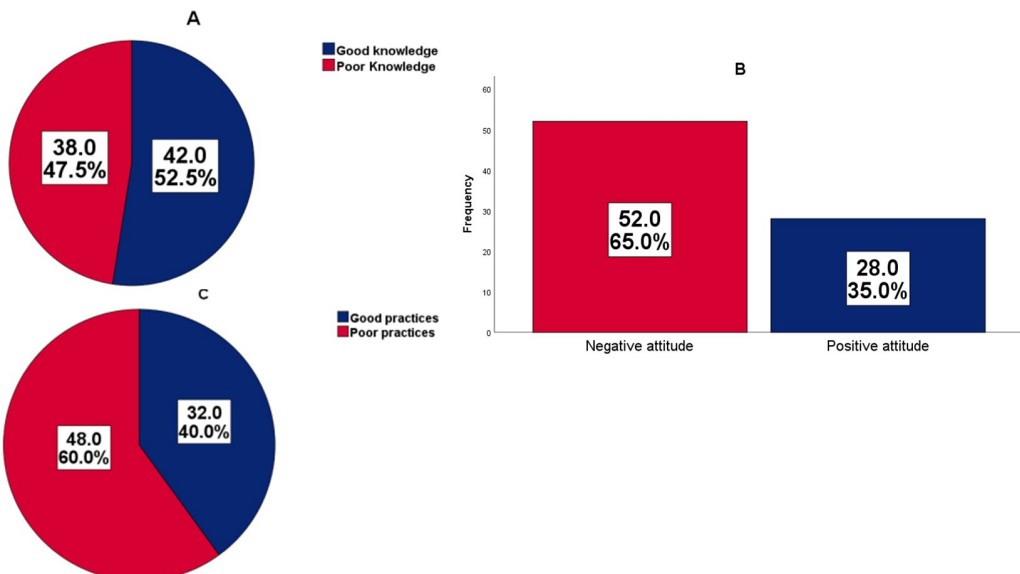

**Fig 1.** Levels of knowledge (A), attitude (B) and practices (C) of CHW on the management and prevention of childhood malaria.

simply as watery stool, others as watery stool > 3 times a day, while others reported that diarrhea is stool mixed with blood.

*"Frequent stools of 3 or more times a day and the stool is usually slimy in nature."* [Male, TIKO]

*"Diarrhea is stool mixed with blood."* [Female, LIMBE]

*"Diarrhea is purging which occurs more than 3 times in a day."* [*Female, BUEA*]

*"The child's eyes will be white and his body will feel weak and he will have fever."* [Female, LIMBE]

*"The child will be dehydrated.*" [Female, TIKO]

CHWs were aware of the methods of preventing childhood diarrhea. Some stated the practice of good hygiene and sanitation, good personal hygiene and others reported exclusive breastfeeding for all babies <6 months of age.

*"A mother should always wash her hands and clean her breast/nipple before breastfeeding the child. For older babies, the mother should give them clean or mineral water for drinking."* [Male, TIKO]

*"Advise the mother to carry out exclusive breastfeeding for six months."* [Female, LIMBE].

## Attitude on management and prevention of childhood diarrhea

The majority of the participants were in agreement that childhood diarrhea cannot be caused by witchcraft (95.0%), disagreed that diarrhea is a disease of the poor (83.8%), and occurs only in Africa (81.3%) (Table 8). Some participants (27.5%) agreed that traditional herbs are better for treating childhood diarrhea, over half (56.3%) agreed that childhood

**Table 7. CHWs' knowledge on the management and prevention of childhood diarrhea.**

| Variable | Frequency (n) | Percentage (%) |
|---|---|---|
| **Definition of diarrhea** | | |
| Passage of loose or Watery stool once a day | 1 | 1.3 |
| Passage of loose or watery stool more than 3 times a day | 69 | 86.3 |
| Passage of hard stool more than three times a day | 5 | 6.3 |
| Passage of hard stool at least once a day | 1 | 1.3 |
| I do not know | 1 | 1.3 |
| "Purge belle" | 3 | 3.8 |
| **Causes diarrhea in children** | | |
| Poor nutrition | 29 | 36.3 |
| Unsafe drinking water | 72 | 90.0 |
| Unsafe fecal disposal | 25 | 31.3 |
| Witchcraft | 0 | 0 |
| Intravenous use of drugs | 2 | 2.5 |
| Teething | 12 | 15.0 |
| I do not know | 0 | 0 |
| **Signs and symptoms of diarrhea** | | |
| Watery stool greater than four days | 25 | 31.3 |
| Greater than four stools per day | 37 | 46.3 |
| Weakness | 34 | 42.5 |
| Sunken eyes | 16 | 20.0 |
| Associated vomiting | 12 | 15.0 |
| Weight loss | 20 | 25.0 |
| Blood in watery stool | 17 | 21.25 |
| Decrease in the quantity of urine | 1 | 1.25 |
| Fever, cold hands and feet | 6 | 7.5 |
| I do not know | 3 | 3.8 |
| **Abstaining from breastmilk prevents childhood diarrhea** | | |
| I strongly agree | 7 | 8.8 |
| I agree | 11 | 13.8 |
| I do not agree | 50 | 62.5 |
| I do not know | 12 | 15.0 |
| **Do you know how to prepare ORS?** | | |
| Yes | 75 | 93.75 |
| No | 5 | 6.25 |
| **Use of ORS as a treatment for childhood diarrhea** | | |
| Never | 1 | 1.3 |
| Sometimes | 22 | 27.5 |
| Always | 48 | 60.0 |
| I do not know | 9 | 11.3 |
| **Zinc supplements as a treatment for childhood diarrhea** | | |
| I strongly agree | 27 | 33.8 |
| I agree | 26 | 32.5 |
| I do not agree | 4 | 5.0 |
| I do not know | 23 | 28.8 |
| **A consequence of childhood diarrhea** | | |
| Loss of consciousness | 7 | 8.8 |
| Weight loss | 55 | 68.8 |

(*Continued*)

**Table 7.** (Continued)

| Variable | Frequency (n) | Percentage (%) |
|---|---|---|
| Blood shortage | 1 | 1.3 |
| Death | 13 | 16.3 |
| I do not know | 4 | 5.0 |
| **Where CHWs advise parents to seek help for childhood diarrhea management** | | |
| Pharmacy | 15 | 18.8 |
| Health center | 66 | 82.5 |
| Hospital | 62 | 77.5 |
| Traditional healer | 1 | 1.3 |
| CHW | 17 | 21.3 |
| Self-medication with herbs | 2 | 2.5 |
| Self-medication with antibiotics | 4 | 5.0 |

diarrhea can be successfully managed at home and 38 (47.5%) reported that a mixture of guava (*Psidium guajava*) leaves and 'masepo' (*Ocimum gratissimum*) leaves are a good home treatment for childhood diarrhea. Most (78.8%) participants were in agreement that teething can cause childhood diarrhea and that if not managed, childhood diarrhea can lead to death (93.8%).

**Table 8. Attitude on the management and prevention of childhood diarrhea.**

| Variable | Frequency (n) | Percentage (%) |
|---|---|---|
| **Childhood diarrhea can be caused by witchcraft** | | |
| I agree | 4 | 5.0 |
| I do not agree | 76 | 95.0 |
| **Diarrhea is a disease of the poor** | | |
| I agree | 13 | 16.3 |
| I do not agree | 67 | 83.8 |
| **Traditional herbs are better for treating children's diarrhea** | | |
| I agree | 22 | 27.5 |
| I do not agree | 58 | 72.5 |
| **Childhood diarrhea occurs only in Africa** | | |
| I agree | 15 | 18.8 |
| I do not agree | 65 | 81.3 |
| **A mixture of guava and 'masepo' leaves is good for treating diarrhea at home** | | |
| I agree | 38 | 47.5 |
| I do not agree | 42 | 52.5 |
| **Diarrhea in a child can lead to death is not managed** | | |
| I agree | 75 | 93.8 |
| I do not agree | 5 | 6.3 |
| **Diarrhea in children can be successfully managed at home** | | |
| I agree | 45 | 56.3 |
| I do not agree | 35 | 43.8 |
| **Teething can cause childhood diarrhea** | | |
| I agree | 63 | 78.8 |
| I do not agree | 17 | 21.3 |

FGD participants agreed that childhood diarrhea can be successfully managed at home without necessarily going to the hospital and most of them recommended the use of traditional herbs including fresh guava leaves, 'masepo' leaves and charcoal mixed with palm oil.

> "Yes, I will recommend that traditional herbs be used because of the unavailability of both ORS and zinc supplements in most health areas and even in health facilities and it is cheap and affordable." [Female, TIKO].

> "I use fresh guava leaves on both children and adults. I can't be precise about the dose, but for adults; they should chew the fresh guava leaves while for children, we boil the leaves and give the child one or two teaspoons three times a day and before you know it the child is fine again." [Male, BUEA].

> "Sometimes I mix charcoal with salt and red oil and give to the child." [Female, TIKO].

### Practices on management and prevention of childhood diarrhea

To diagnose diarrhea in children, most participants (66.25%) used the level of dehydration in the child while (41.25%) used the mid-upper arm circumference (Table 9). The majority (51, 63.8%) knew how to prepare ORS and 64 (80.0%) administered ORS as a treatment for childhood diarrhea. Over half (53.8%) knew how to prepare salt sugar solution (SSS), sixty-two (77.5%) used SSS alongside zinc supplements (49, 61.35%) and only 16 (20%) administered fluids more than usual during diarrheal episodes. Alternative treatments administered to children with diarrhea were; squeezed guava leaves juice (35.0%), palm kenel and guava leaves (12.5%), charcoal, oil and salt (7.5%), and palm oil (5.0%). Most CHWs (72.5%) advised mothers to practice exclusive breastfeeding for their babies.

FGD participants also had varied opinions on the preparation of ORS and SSS for the management of childhood diarrhea.

> "The ORS comes in its packet with tablets. So, we add one sachet of ORS into 1 litter of clean water." [Female, BUEA]

> "We add 8–10 cubes of sugar into 1L of clean water with half a teaspoon of salt." [Female, TIKO]

> "For one liter of water, I add ½ teaspoon of salt with 6 cubes of sugar inside." [Female, LIMBE]

> "By mixing 1 teaspoon of salt in 1.5 liters of water and adding 8 cubes of sugar." [Male, LIMBE]

> "One level teaspoon of salt, five cubes of sugar in one liter of water then you shake until the mixture is uniform." [Male, BUEA]

Some administered ORS and zinc supplements while others administered herbs.
FGD participants reported they manage childhood diarrhea as follows:

> "I prepare and give the child oral rehydration salt solution and then observe the child." [Female, TIKO]

> "Sometimes I give fresh (young) guava leaves and I advise children to chew." (Male, TIKO)

**Table 9. Practices on the management and prevention of childhood diarrhea.**

| Variable | Frequency (n) | Percentage (%) |
|---|---|---|
| **Diagnosis of childhood diarrhea** | | |
| Measuring the mid-upper arm circumference | 33 | 41.3 |
| Measuring the child's height | 11 | 13.8 |
| Asking the child's age | 6 | 7.5 |
| Noting the level of dehydration in the child | 53 | 66.3 |
| I don't know | 31 | 38.8 |
| **Administration of ORS** | | |
| Yes | 64 | 80.0 |
| No | 16 | 20.0 |
| **How will you prepare an ORS solution** | | |
| Put 1 pack of ORS in 1 liter of water | 51 | 63.8 |
| Put 1 pack of ORS in 1.5 liters of water | 20 | 25.0 |
| Put 1 pack of ORS in water | 4 | 5.0 |
| I do not know | 3 | 3.8 |
| I have forgotten | 2 | 2.5 |
| **Use zinc supplements to treat childhood diarrhea** | | |
| Yes | 49 | 61.3 |
| No | 16 | 20.0 |
| I do not know | 15 | 18.8 |
| **Use salt sugar solution as a treatment for childhood diarrhea** | | |
| Yes | 62 | 77.5 |
| No | 18 | 22.5 |
| **Items used to prepare salt sugar solution** | | |
| 1teaspoon of salt, and 10 level teaspoons of sugar in 1 liter of clean water | 15 | 18.8 |
| 0.5 teaspoon of salt, 5 cubes of sugar in 1 liter of clean water | 35 | 43.8 |
| 1 teaspoon of salt, 5 cubes of sugar in 1.5 liters of clean water | 9 | 11.3 |
| 0.5 teaspoon of salt, 6 level teaspoons of sugar in 1 liter of clean water | 8 | 10.0 |
| I do not know | 13 | 16.25 |
| **Administration of fluids during diarrhea episodes** | | |
| Less than usual | 14 | 17.5 |
| Same as usual | 38 | 47.5 |
| More than usual | 16 | 20.0 |
| I do not know | 12 | 15.0 |
| **Advisable feeding methods for babies** | | |
| Formula milk | 15 | 18.75 |
| No feeding, just mineral water | 6 | 7.5 |
| Exclusive breastfeeding | 58 | 72.5 |
| Juice or yogurt | 1 | 1.25 |
| Others | 0 | 0.0 |
| **Alternative treatment for childhood diarrhea** | | |
| Charcoal, oil and salt | 6 | 7.5 |
| Palm oil | 4 | 5.0 |
| Squeezed guava leave juice | 28 | 35.0 |
| Palm kenel and guava leaves | 10 | 12.5 |
| None of the above | 32 | 40.0 |

## Correlation of CHWs scores on KAP on childhood diarrhea management and prevention

The mean knowledge score (± SD) on childhood diarrhea management and prevention was 18.25 (±2.568), attitude 6.03 (±1.091) and practices 25.45 (±4.8). Bivariate analysis showed that knowledge had a significant positive association with attitude (r = 0.265, p = 0.018) and practices (r = 0.329, p = 0.003) (Table 10).

The study showed a significant association between the level of education (p = 0.028) and health district (p = 0.026) with practices on childhood diarrhea management and prevention. The majority (57.5%) of CHWs had poor knowledge, negative attitude (70.0%) but carried out good practices (82.5%) on childhood diarrhea management and prevention (Fig 2).

## Discussion

This study revealed that 52.5% of CHWs had good knowledge on childhood malaria management and prevention though only 65.0% correctly identified fever, headache, loss of appetite and body pains as signs and symptoms of simple malaria. These findings however contradict the reports of a study in Bamenda, Cameroon where 97.8% of CHWs knew mosquitoes transmit malaria and 95.5% knew the most common signs/symptoms of malaria [18]. In addition, a study in Rwanda showed 89.2% of CHWs knew mosquito as the vector that transmits malaria to humans and 73.8% knew the common signs and symptoms of malaria [21]. Likewise, temperature greater than 37°C and loss of consciousness was recognized by just 10.0% of CHWs as signs and symptoms of severe malaria, and only 30% agreed RDT and treatment with antimalarial medications should be done for every suspected case of childhood malaria. Thirty-four (42.5%) CHWs knew ACT as the first-line treatment for malaria in Cameroon while some CHWs (8.75%) did not know the first-line treatment for malaria in Cameroon. This value is lower than what was observed in Nigeria where a higher percentage (76.0%) of CHWs made use of ACT [25]. A possible reason for the poor knowledge registered by CHWs could be the lack of involvement of all CHWs in training, as reported in the FGDs. In Cameroon, even though malaria has been a major public health problem in the country for decades, CHWs are trained in specific disease control programs including malaria, vaccine-preventable diseases and NTDs like leishmaniasis, onchocerciasis and schistosomiasis. Training is usually done by staff from the Ministry of Public Health, and local and international non-governmental organizations (NGOs), often before health campaigns and MDA programs and not regularly. In this study, not all CHWs were trained to manage childhood malaria and diarrhea. Some CHWs in the FGDs reported that the management of childhood malaria and childhood diarrhea is not in their minimum package of activities, while most reported that only a few members from each health district were selected and trained to manage these diseases. The

**Table 10. Descriptive statistics for KAPs of CHWs regarding the management of childhood diarrhea.**

| Variable | Total | Minimum | Maximum | Mean | Standard deviation | Spearman Correlation and $X^2$ value | | |
|---|---|---|---|---|---|---|---|---|
| | | | | | | Knowledge | Attitude | Practices |
| **Knowledge Score for diarrhea management** | 30 | 13 | 24 | 18. 25 | 2.568 | r = 1.00 | r = 0.265* | r = 0.329** |
| **Attitude Score for diarrhea management** | 11 | 2 | 9 | 6.03 | 1.091 | r = 0.265* p-value = 0.018 | r = 1.000 | r = 0.158 |
| **Practices Score for diarrhea management** | 38 | 14 | 35 | 25.45 | 4.8 | r = 0.329** p-value = 0.003 | r = 0.158 | r = 1.000 |

**Correlation is significant at the 0.01 level (2-tailed).

*Correlation is significant at the 0.05 level (2-tailed).

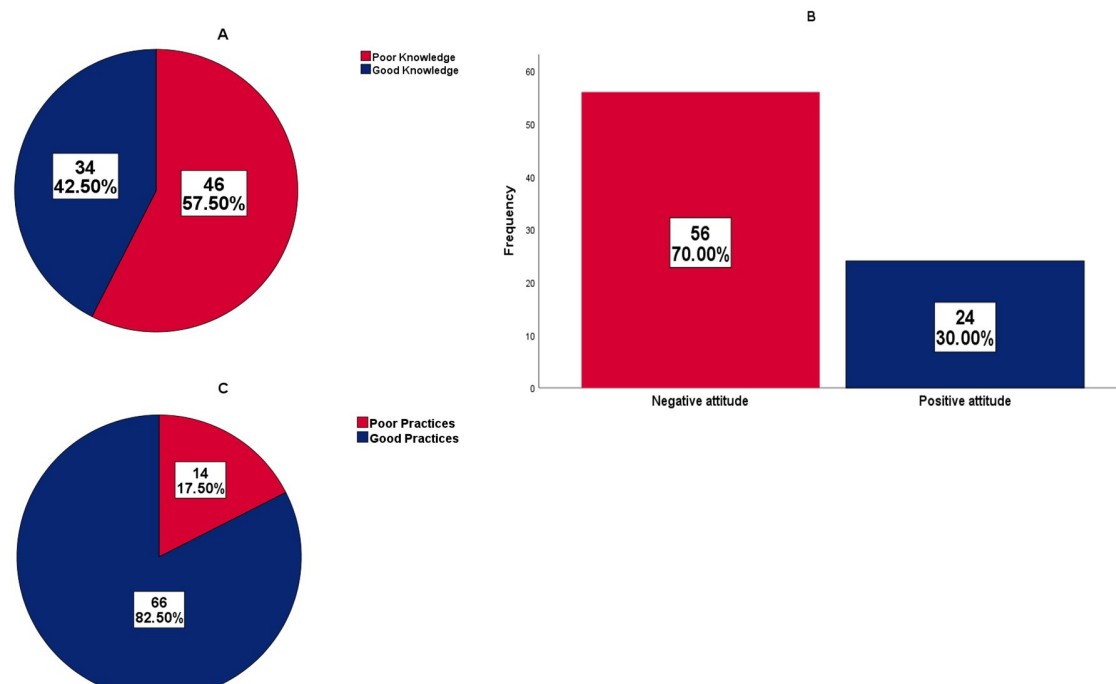

**Fig 2.** CHWs' level of knowledge (A), attitude (B) and practices (C) on the management and prevention of childhood diarrhea.

minimum package of activities cuts across programs within which CHWs play developmental, preventive and curative roles that include; home visits, awareness building, preventive care, first aid clinical services, follow-up and timely referrals of severe cases. This could be a possible demotivation to the greater proportion of CHWs who are not trained. Hence, there is a need for the involvement of all CHWs in trainings, to improve their knowledge and stimulate behavior change.

The CHWs had negative attitude regarding childhood malaria management and prevention (65.0%). Some of the study participants (7.5%) agreed that severe childhood malaria can be caused by witchcraft, 75.0% agreed that a mixture of boiled paw-paw, mango leaves and fever grass (lemon grass) is a good home treatment for malaria, and 46.25% were in agreement that, side pain is a cause of malaria in children. These values are lower than those reported in Bamenda where 97.0% of CHWs had good attitude towards malaria management and prevention [19]. The negative attitude of CHWs could have resulted from the lack of resources in the study settings as well as poor knowledge resulting from the lack of training of some CHWs as stated during the FGDs. Cameroon being a resource-limited country, fails to constantly supply RDT test strips, antimalarial drugs, and paracetamol to CHWs for the management of childhood malaria in the community. When available in pharmacies these materials are expensive and not subsidized hence CHWs may result in the use of traditional herbs, which are readily available, accessible and free to get. This further reiterates the need for training to change the attitudes of these CHWs as they may be passing wrong information to the communities they serve. However, 97.5% of CHWs believed malaria is a life-threatening disease, this is similar to the findings of reported in Bamenda 97.0%) and in Uganda (97.1%) [19,22].

More than half (60.0%) of CHWs carried out poor practices on the management and prevention of childhood malaria; 51.25% had never given health education on hygiene and sanitation to community members. Only 57.5% did RDT for malaria, 16.25% did not know the

number of doses to administer for malaria treatment in a day, and 75.0% of CHWs agreed that a mixture of boiled paw-paw, mango leaves and fever grass (lemon grass) is a good home treatment for childhood malaria. Similar to this study, CHWs in Senegal failed to administer RDT to 24.3% of malaria-symptomatic patients [31]. This is contrary to what was observed in Nigeria where 83.7% of the participants carried out good practices in which participants wrote instructions and counseled patients on how to use antimalarial drugs and their expected side effects, and advised them to always sleep under mosquito nets [23]. Another study in Senegal reported that CHW treated 98.6% of all RDT-positive cases of malaria (29% were children < 5 years) of whom 77% were administered ACT [24]. This finding is similar to a study in Nigeria where 37.0% of the CHWs preferred to use herbs in treating malaria [23]. Thus, CHWs in this study does not fully understand hygiene as a preventive measure for malaria. Though they recognized bed net as a preventive strategy, most (82.5%) did not have LLINs for distribution. CHWs should be provided with these nets for distribution.

In general, CHWs (57.5%) had inadequate knowledge on childhood diarrhea management; only 60% always administered ORS as a treatment for childhood diarrhea and 28.75% did not know that zinc supplement is used in treating childhood diarrhea. These results are similar to what was observed in Kenya, where all the CHWs knew that diarrhea should be treated with ORS, but none of them knew that all children with diarrhea should receive zinc [16]. From the results of this study, it can be suggested that the high prevalence of diarrhea in children <5 years in the study area as previously reported [8] and also as observed in this study, could be associated with CHWs' poor knowledge of diarrhea management. This underscores the need for training to educate CHWs on the management of childhood diarrhea.

CHWs (70.0%) had negative attitude, from the FGDs participants agreed that childhood diarrhea can be successfully managed at home without necessarily going to the hospital (56.25%) and recommended the use of traditional herbs including fresh guava (*Psidium guajava*) leaves, 'masepo' (*Ocimum gratissimum*) and charcoal mix with palm oil (47.5%) as good home treatment for childhood diarrhea. From these findings, probably, the high prevalence of childhood diarrhea in the study area [8] is as a result of the negative attitude CHWs have toward the management and prevention of childhood diarrhea, which justifies the need for more training to bring about behavior change.

The study reported participants (82.5%) carried out good practices for the management and prevention of childhood diarrhea. However, CHW (17.5%) reported the use of unconventional practices including administering less fluid than usual (17.0%), fluid same as usual (47.5%) and squeezed guava leaves (35.0%) as a treatment for childhood diarrhea. Participants in the FGDs also repeated similar practices. Such unconventional practices have also been reported in Indonesia where *Psidium guajava* leaves, curcumic (turmeric), and tea, occasionally mixed with salt were used as traditional diarrheal treatment [32]. On the contrary, a study in Senegal reported that CHW appropriately used ORS and zinc to manage 99.6% of all diarrhea cases [24]. These findings further stretch on the importance of continuously improving training programs as well as organizing evaluation activities to improve the practices of CHWs since they represent the health system in the community.

Bivariate correlation showed that knowledge on children's malaria significantly contributed to attitude positively, and attitude significantly correlates with practices. This finding stretches the importance of developing good prevention and management programs to raise awareness among CHWs which will then motivate behavior change.

For the management of childhood diarrhea, the study found a significant positive correlation between knowledge with attitude and knowledge with practices. Also, there was a significant association between the level of education and health district with practices on childhood diarrhea management and prevention. Similar to our findings, Wanduru *et al.* reported

secondary-level education to be positively associated with good performance in the management of diarrhea in addition to meeting with supervisors the previous month [22]. Another study in South Sudan showed significant associations between: knowledge and education; practices and education; age and attitude; diarrhea and income [33]. The findings of our research study suggest that CHWs' level of education should be greatly considered when selecting CHWs since more educated CHWs were likely to carry out better practices when managing childhood diarrhea. In addition, CHWs generally had inadequate knowledge regarding the management of diarrhea. Hence more training needs to be done to raise their awareness about the disease.

## Conclusion

The study showed that some CHWs are trained to manage childhood malaria as seen in their overall knowledge of the disease. However, the facilities necessary for disease management are not available; this is reflected on their attitude and practices. The overall poor knowledge and attitude on childhood diarrhea management show that CHWs have not been effectively trained to manage the disease. However, they have background knowledge from their everyday life where they use traditional herbs in the treatment of the diarrhea. Therefore, there is a need for further training to bridge these gaps. The study recommends that local NGOs and the government should provide ACT, ORS and mosquito nets to reinforce the activities of community health workers.

## Limitations of the study

This study is limited in methodology. Even though the FGD guide had questions on previous training on childhood malaria and diarrhea the KAP survey questionnaire did not include this specific question. Another limitation is that the questionnaire did not include a specific question on previous training on childhood malaria and diarrhea.

## Supporting information

**S1 Table. Table of themes.**
(DOCX)

**S1 Data.**
(XLSX)

**S1 Questionnaire.**
(DOCX)

**S1 Text. Focus group discussion interview guide.**
(DOCX)

## Acknowledgments

The authors are grateful to the community health workers of Fako Division for taking part in this study.

## Author Contributions

**Conceptualization:** Ndum Naomi Chi, Raymond Babila Nyasa.

**Formal analysis:** Ndum Naomi Chi, Raymond Babila Nyasa, Akoachere Jane-Francis.

**Investigation:** Ndum Naomi Chi, Akoachere Jane-Francis.

**Methodology:** Ndum Naomi Chi, Raymond Babila Nyasa, Akoachere Jane-Francis.

**Supervision:** Raymond Babila Nyasa, Akoachere Jane-Francis.

**Writing – original draft:** Ndum Naomi Chi, Raymond Babila Nyasa, Akoachere Jane-Francis.

**Writing – review & editing:** Raymond Babila Nyasa, Akoachere Jane-Francis.

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
