## [Decision Letter · Decision Letter 0]

24 May 2022

PGPH-D-22-00391

Knowledge attitude and practices of community health workers on the management and prevention of childhood malaria and diarrhea in Fako Division, South West Region, Cameroon

Dear Dr. Naomi,

Thank you for submitting your manuscript to PLOS Global Public Health. After careful consideration, we feel that it has merit but does not fully meet PLOS Global Public Health’s publication criteria as it currently stands. Therefore, we invite you to submit a revised version of the manuscript that addresses the points raised during the review process.

Please submit your revised manuscript by . If you will need more time than this to complete your revisions, please reply to this message or contact the journal office at globalpubhealth@plos.org. Please include the following items when submitting your revised manuscript:

We look forward to receiving your revised manuscript.

Kind regards,

Guglielmo Campus, Ph.D DDS

Academic Editor

Journal Requirements:

1. Please provide an Author Summary. This should appear in your manuscript between the Abstract (if applicable) and the Introduction, and should be 150–200 words long. The aim should be to make your findings accessible to a wide audience that includes both scientists and non-scientists. Sample summaries can be found on our website under Submission Guidelines: https://journals.plos.org/globalpublichealth/s/submission-guidelines#loc-parts-of-a-submission

Alternative link: http://journals.plos.org/ploscompbiol/s/submission-guidelines#loc-author-summary

Additional Editor Comments (if provided):

Reviewers' comments:

Reviewer's Responses to Questions

**Comments to the Author**

1. Does this manuscript meet PLOS Global Public Health’s publication criteria? Is the manuscript technically sound, and do the data support the conclusions? The manuscript must describe methodologically and ethically rigorous research with conclusions that are appropriately drawn based on the data presented.

Reviewer #1: Partly

Reviewer #2: Yes

2. Has the statistical analysis been performed appropriately and rigorously?

Reviewer #1: No

Reviewer #2: Yes

3. Have the authors made all data underlying the findings in their manuscript fully available (please refer to the Data Availability Statement at the start of the manuscript PDF file)?

Reviewer #1: No

Reviewer #2: Yes

4. Is the manuscript presented in an intelligible fashion and written in standard English?

Reviewer #1: Yes

Reviewer #2: Yes

5. Review Comments to the Author

Reviewer #1: ABSTRACT:

There is no conclusion section for the abstract. Would divide Line 39 into Conclusion section

INTRO

Line 58: That sentence is a little confusing. It would be good to clarify the definition of ‘the world’s 18th country’. Is that a ranking? Reference #5 is also not appropriately written in the References Section so reader is unable to refer to the primary source of information to get a clearer understanding of the sentence.

Line 61: Can remove ‘the diseases’ in the sentence as the listed diseases are self-explanatory.

Line 84: Assuming KAP refers to ‘Knowledge attitude and Practices’ but this is not alluded to in any section of the INTRODUCTION. This needs to be fully spelled out initially before abbreviating. Although there is a list of abbreviations at the end, it would be helpful to the reader to see the full word in the body of the paper initially and then made into an abbreviation when used again. Would check that this occurs for each abbreviation

Last paragraph references other studies that have done work on community healthcare workers and what they have reported. There indeed is also the lack of literature on childhood malaria and diarrhea in Cameroon. However, authors have not stated the AIMS of this study or the PURPOSE for conducting this study. There is also not a clear hypothesis for the research question being asked. The last paragraph should clarify the research question

METHODS:

Line 113: refer to HD before using the abbreviation

Study Design seems to be mixed-methods with quantitative portion described in analysis but qualitative component no clarified. This should be revisited by the authors to determine the appropriate methodology and study design as this seems to be more like a cross-sectional mixed-methods study

Line 116: good description for why this health district was left out

Line 118: Should provide description of Inclusion criteria? The subsection for “Criteria for Inclusion/Exclusion” does not detail the inclusion and exclusion criteria in detail. Would revisit this to help reader understand how participants were chosen

What is the total number of participants in the study? What is the sample size or the sampling procedure for the study? Recruitment? How were certain participants excluded and why? The methods should go more into detail about the criteria used. This could be expressed as a figure as well

Line 156: there should be reference for the data analysis program for SPSS version 25

Quantitative analysis is described but the data from the focus groupsis not explained on how themes were created from the discussion. Examples of types of analysies include reflexive thematic analysis? Deductive analysis? Was the thematic analysis done by one author or multiple? The methods should include the analysis of the Qualitative portion as these responses in the focus groups are a large part of the Results section

Lines 162-163: would be beneficial to understand the scores if described further in detail

Would revisit the analysis and input p-values when discussing the associations between outcome and variables. The analyses needs more description

RESULTS

Sociodemographic characteristics would be well displayed as a table rather than in a paragraph. This would help the reader easily refer to this and also improve word count.

First sentence seems incomplete: “Only of the 143 CHWs work in Fako…” Unsure if there is supposed to be a number after the first word.

Line 185-187: the first sentence is confusing. “composed of 9 participants each 11 participants were involved.” Would clarify more with this statement

Would recommend either shortening the statements added from the focus groups or make a supplemental figure for the responses so reader can refer to them. The addition of the full responses prolongs the wording of the paper and can be confusing at times.

DISCUSSION:

Paragraph starting on line 539, first sentence is a repeat of the prior paragraph.

This should be cross-sectional mixed methods

OVERALL: Manuscript will need to provide further information regarding its methodology and analyses. This was noted to be a cross-sectional study in the Methods section, however, the addition of qualitative data and themes according to the results section will need to be described further. The presentation of the data in the Results section should be further expanded as the body of the Results section is very wordy. The use of demographic figures or tables and other forms of representation of the qualitative data should be explored.

Reviewer #2: Summary and Overall recommendation

The main research question is whether there is a correlation between the level of knowledge, attitudes and practices of community healthcare workers in the prevention and management of childhood malaria and diarrhea in the Fako Division , South West Region, Cameroon. The study results and data analysis show a positive correlation between a higher level of knowledge which influences a more positive attitude and use of the correct practices in the prevention and management of childhood malaria and diarrhea in line with the country’s and WHO guidelines (lines 534-541,page 37). The conclusion of the study is that amongst the study population , more than 50% of the 80CHWs who participated in the study had good knowledge and more than 60% had negative attitudes and poor practices on childhood malaria management and prevention. The conclusion on childhood diarrhea management and prevention showed similar results where more than 50% of the CHWs had poor knowledge, 75% had negative attitudes and more than 75% used good practices. There were interesting conclusions on the preference for traditional herbal concoctions in the treatment of both childhood malaria (75%) and diarrhea (33%) which is in line with the correlation findings on the knowledge, practices and attitudes discussed above. The research fits in with existing literature in the region with references drawn from Nigeria, Rwanda and Kenya.

Strengths of the manuscript are on the study design, data tools, collection and analysis of the data and description of the results. The selection of the Chi Square and Spearman Correlation tests was appropriate. The study setting discussion section could be strengthened with more details as highlighted in the section on major issues below. Overall this is a good study on Knowledge, Attitude and Practices of CHWs on management and prevention of childhood malaria and diarrhea as a first of its kind in Cameroon and is recommended for publication following requested revisions

Major Issues

I. Study Setting: Include a description of the community health care system in Cameroon, in particular how the CHWs are linked to the health centers, health areas and district. Also include the number of health centers that the CHWs serve or refer patients to in the 3 health districts selected and whether these are only public or private facilities. This is important in creating a better context towards understanding the CHWs operating environment .

II. Discussion:

• Line 474, kindly give examples of the other disease control programs that CHWs are trained on

• Line 476, kindly include more detail on what the minimum package of activities are for CHWs in the region

• Line 487, discussion only focuses lack of resources on negative attitudes by CHWs towards malaria management and prevention; what about the influence of low knowledge or lack of access to trainings as mentioned earlier

• Line 499 to 501, the comparison with Nigeria on the practice of using herbs in treatment of malaria shows quite a significant difference with Cameroon at 75% and Nigeria at 37% recommend that authors look for a country/ study with statistics with a closer similarity in results for better comparison

III. Literature references can be strengthened by drawing more examples from studies in the West Africa region as opposed to East Africa as the endemicities and cultural practices vary slightly. E.g. line 529 pulls a reference from Indonesia in South East Asia.

6. PLOS authors have the option to publish the peer review history of their article (what does this mean?). If published, this will include your full peer review and any attached files.

**Do you want your identity to be public for this peer review?** For information about this choice, including consent withdrawal, please see our Privacy Policy.

Reviewer #1: No

Reviewer #2: No

---

## [Decision Letter · Decision Letter 1]

22 Sep 2022

PGPH-D-22-00391R1

Knowledge attitude and practices of community health workers on the management and prevention of childhood malaria and diarrhea in Fako Division, South West Region, Cameroon

Dear Dr. Naomi,

Thank you for submitting your manuscript to PLOS Global Public Health. After careful consideration, we feel that it has merit but does not fully meet PLOS Global Public Health’s publication criteria as it currently stands. Therefore, we invite you to submit a revised version of the manuscript that addresses the points raised during the review process.

We look forward to receiving your revised manuscript.

Kind regards,

Sarah Elizabeth Brewer, PhD

Academic Editor

Journal Requirements:

Additional Editor Comments (if provided):

While revisions have made this manuscript stronger, the reviewers raised a number of remaining issues that will need to be addressed. In addition to the reviewers comments, please review the COREQ checklist for reporting your qualitative findings.  Also consider whether you have streamlined your reporting on both malaria and diarrhea so the findings are clear and concise.  

Reviewers' comments:

Reviewer's Responses to Questions

**Comments to the Author**

1. If the authors have adequately addressed your comments raised in a previous round of review and you feel that this manuscript is now acceptable for publication, you may indicate that here to bypass the “Comments to the Author” section, enter your conflict of interest statement in the “Confidential to Editor” section, and submit your "Accept" recommendation.

Reviewer #3: All comments have been addressed

Reviewer #4: (No Response)

Reviewer #5: All comments have been addressed

Reviewer #6: All comments have been addressed

2. Does this manuscript meet PLOS Global Public Health’s publication criteria? Is the manuscript technically sound, and do the data support the conclusions? The manuscript must describe methodologically and ethically rigorous research with conclusions that are appropriately drawn based on the data presented.

Reviewer #3: Yes

Reviewer #4: Partly

Reviewer #5: Yes

Reviewer #6: Yes

3. Has the statistical analysis been performed appropriately and rigorously?

Reviewer #3: Yes

Reviewer #4: No

Reviewer #5: I don't know

Reviewer #6: Yes

4. Have the authors made all data underlying the findings in their manuscript fully available (please refer to the Data Availability Statement at the start of the manuscript PDF file)?

Reviewer #3: Yes

Reviewer #4: Yes

Reviewer #5: Yes

Reviewer #6: Yes

5. Is the manuscript presented in an intelligible fashion and written in standard English?

Reviewer #3: No

Reviewer #4: Yes

Reviewer #5: Yes

Reviewer #6: Yes

6. Review Comments to the Author

Reviewer #3: This comprehensive manuscript describes the level of knowledge, attitude and practices on childhood malaria and diarrhoea among 80 CHWs in southwestern Cameroon. Overall, the study shows that the studied CHWs had poor level of knowledge and negative attitude towards childhood malaria and diarrhoea. Also, the study reports a positive correlation between a higher level of knowledge and more positive attitude and higher level of appropriate related practices.

I've been given this manuscript as a revised (R1) version and I noticed that two reviewers have already seen this in its original form. I’ve read the reviewer’s comments and authors’ responses.

Based on my evaluation, the manuscript is interesting, comprehensive, and reports important results that can be useful for policymakers in the targeted communities. This version of the manuscript has been improved; however, it still needs more improvement before it can be acceptable for publication. Specific comments are provided in the attached file.

Reviewer #4: Many thanks for the opportunity to review this manuscript. This manuscript could add to the literature on childhood malaria and diarrhea control in Sub-Saharan Africa. The use of mixed methods helps enrich the study. The authors should consider the following suggestions for improving the manuscript:

Please see the attachment.

Reviewer #5: (No Response)

Reviewer #6: The paper is much improved with the responses to the initial questions. However, I still have 2 suggestions to improve the paper:

1. Please do a review of all the tables as some of the content had simple editorial errors which need to be corrected.

2. The conclusion was weak. It was a summary of the findings but with no actual conclusion. What does this study say about the CHW program? Does it offer any suggestions for areas to improve? Etc. Conclusions should go one step beyond the findings.

7. PLOS authors have the option to publish the peer review history of their article (what does this mean?). If published, this will include your full peer review and any attached files.

**Do you want your identity to be public for this peer review?** For information about this choice, including consent withdrawal, please see our Privacy Policy.

Reviewer #3: **Yes: **Hesham M. Al-Mekhlafi

Reviewer #4: No

Reviewer #5: No

Reviewer #6: No

---

## [Editor Report · Decision Letter 2]

24 Jan 2023

Knowledge, attitude and practices of community health workers on managing and preventing childhood malaria and diarrhea in Fako Division, South West Region, Cameroon; A mixed method study

PGPH-D-22-00391R2

Dear Ms Naomi,

We are pleased to inform you that your manuscript 'Knowledge, attitude and practices of community health workers on managing and preventing childhood malaria and diarrhea in Fako Division, South West Region, Cameroon; A mixed method study' has been provisionally accepted for publication in PLOS Global Public Health.

Best regards,

Sarah Elizabeth Brewer, PhD

Academic Editor